# Serum Neurofilament Light Chain Levels are Associated with Lower Thalamic Perfusion in Multiple Sclerosis

**DOI:** 10.3390/diagnostics10090685

**Published:** 2020-09-11

**Authors:** Dejan Jakimovski, Niels Bergsland, Michael G. Dwyer, Deepa P. Ramasamy, Murali Ramanathan, Bianca Weinstock-Guttman, Robert Zivadinov

**Affiliations:** 1Buffalo Neuroimaging Analysis Center (BNAC), Department of Neurology, Jacobs School of Medicine and Biomedical Sciences, University at Buffalo, State University of New York, Buffalo, NY 14203, USA; npbergsland@bnac.net (N.B.); mgdwyer@bnac.net (M.G.D.); dramasamy@bnac.net (D.P.R.); rzivadinov@bnac.net (R.Z.); 2IRCCS, Fondazione Don Carlo Gnocchi ONLUS, 20121 Milan, Italy; 3School of Pharmacy, University at Buffalo, State University of New York, Buffalo, NY 14214, USA; murali@buffalo.edu; 4Jacobs Comprehensive MS Treatment and Research Center, Department of Neurology, Jacobs School of Medicine and Biomedical Sciences, University at Buffalo, State University of New York, Buffalo, NY 14203, USA; bw8@buffalo.edu; 5Center for Biomedical Imaging at Clinical Translational Science Institute, University at Buffalo, State University of New York, Buffalo, NY 14203, USA

**Keywords:** multiple sclerosis, perfusion, serum neurofilament light chain, DSC–PWI, hypoperfusion, biomarkers

## Abstract

Both perfusion-weighted imaging (PWI) measures and serum neurofilament light (sNfL) chain levels have been independently associated with disability in multiple sclerosis (MS) patients. This study aimed to determine whether these measures are correlated to each other or independently describe different MS processes. For this purpose, 3T MRI dynamic susceptibility contrast (DSC)–PWI and single-molecule assay (Simoa)-based sNfL methods were utilized when investigating 86 MS patients. The perfusion measures of mean transit time (MTT), cerebral blood volume (CBV), and cerebral blood flow (CBF) were derived for the normal-appearing whole brain (NAWB), the normal-appearing white matter (NAWM), the gray matter (GM), the deep GM (DGM), and the thalamus. The normalized CBV and CBF (nCBV and nCBV) were calculated by dividing by the corresponding NAWM measure. Age- and sex-adjusted linear regression models were used to determine associations between the DSC–PWI and sNfL results. False discovery rate (FDR)-adjusted *p*-values < 0.05 were considered statistically significant. A greater age and thalamic MTT were independently associated with higher sNfL levels (*p* < 0.001 and *p* = 0.011) and explained 36.9% of sNfL level variance. NAWM MTT association with sNfL levels did not survive the FDR correction. In similar models, a lower thalamic nCBF and nCBV were both associated with greater sNfL levels (*p* < 0.001 and *p* = 0.022), explaining 37.8% and 44.7% of the variance, respectively. In conclusion, higher sNfL levels were associated with lower thalamic perfusion.

## 1. Introduction

Multiple sclerosis (MS) is an inflammatory disease of the central nervous system (CNS) that presents as variable levels of demyelination, neurodegeneration, and axonal loss [1]. Moreover, MS is linked with heterogeneous susceptibility and progression processes that involve genetics, environmental factors, and cardiovascular and perfusion deficits. [2] These intertwined pathophysiological processes lead to intermittent neurological impairments that progressively accumulate and result in physical and cognitive disability. To provide timely disease treatment, multiple methods for early detection have been proposed. They range from traditional magnetic resonance imaging (MRI) measures to emerging serum-derived disease markers and non-conventional MRI sequences. Given that each disease biomarker may specifically describe different pathophysiological processes, it is important to determine their divergent or concurrent utility [3,4].

The measurement of neurofilament light chain (NfL) levels is emerging as a promising biomarker for the monitoring and management of neurodegenerative diseases, such as Alzheimer’s disease, MS, and Parkinson’s disease. [5] The NfL is a constituent of the axonal cytoskeleton and its concentrations in bodily fluids reflect the degree of ongoing axonal pathology [5]. Because of the analytical limitations of previous generation assays, NfL levels were originally measurable only in samples with high concentrations, e.g., cerebrospinal fluid (CSF) [6]. However, with the development of fourth-generation single-cell array assays, such as Simoa, it is possible to measure concentrations in serum (sNfL). As such, sNfL has been utilized in various MS studies and successfully verified as a reliable proxy of axonal neuroinflammatory and neurodegenerative pathology [7].

Brain perfusion is an integral part of a larger homeostatic mechanism that allows for the sufficient delivery of oxygen and nutrients necessary for optimal neuronal functioning. In particular, perfusion deficits may result in insufficient MS lesion repair and exacerbate mitochondrial failure. This lack of lesion repair may contribute to a greater accumulation of severe MS pathology (T1 hypointensities and black holes) in hypoperfused brain regions [8]. Perfusion deficits can be monitored in vivo with specific MRI protocols, such as dynamic contrast-enhanced (DCE), dynamic susceptibility contrast (DSC), and arterial spin labeling (ASL) acquisitions [9]. Discrepant results have been published regarding perfusion alterations in MS. Some studies do demonstrate a significant decrease in cerebral perfusion and correlations with both physical and cognitive MS changes [10]. These positive associations were reported in the absence of or in addition to existing structural pathology [10,11]. In particular, one study demonstrated that the only factor differentiating cognitively impaired from non-impaired MS patients was lower cortical perfusion, which argues against associations between neuronal loss and the resulting hypoperfusion [10]. This finding was also suggested in an MRI-based MS study that demonstrated that hypoperfusion and cortical atrophy can occur in different areas of the brain and do not necessarily depend on each other [12].

With this background, the question remains regarding whether perfusion-weighted imaging (PWI)-based measures and sNfL levels describe the same or independent pathophysiological MS processes. We aimed to determine whether the perfusion measures in MS patients would be related to the sNfL levels or are independent.

## 2. Materials and Methods

### 2.1. Study Population

The study population was derived from a larger cardiovascular, environmental, and genetic study in MS (CEG-MS) [13,14]. The inclusion criteria consisted of (1) a diagnosis of MS per McDonald’s 2010 criteria [15] or clinically isolated syndrome (CIS), (2) aged 18–75 years old, (3) availability of the DSC–PWI sequence as part of the MRI protocol, and (4) availability of a blood sample for sNfL level quantification. The blood sample was acquired at the time of the MRI and within 30 days of the clinical examination. The exclusion criteria consisted of (1) being pregnant or a nursing mother, (2) the presence of other major neurological disorders, (3) the presence of a cerebral vascular pathology and malformations that may affect the PWI measures, and (4) a relapse or having received intravenous corticosteroids within the last 30 days at the time of the MRI examination. An experienced neurologist examined the patients and the Expanded Disability Status Scale (EDSS) scores were calculated [16]. The MS patients were classified into relapsing-remitting MS (RRMS) and progressive MS (PMS). Due to sample size limitations, primary-progressive MS (PPMS) and secondary-progressive MS (SPMS) patients were merged into a single PMS group. A standardized questionnaire was used to collect additional data regarding the patient’s body mass index (BMI) and disease-modifying treatment (DMT). The study was approved by the University at Buffalo Institutional Review Board (IRB ID 030-603069, last update 2/26/2020) and all patients signed a written informed consent form.

### 2.2. MRI Acquisition and Analysis

All study subjects underwent a 3T MRI protocol with a Signa Excite HD 12 Twin-Speed scanner (GE, Milwaukee, WI, USA) equipped with an eight-channel head and neck coil. For this analysis, a high-resolution 3-dimensional (3D) T1-weighted imaging (WI) sequence, a 2D fluid-attenuated inverse recovery (FLAIR)-T2 sequence, and a DSC–PWI sequence were utilized. In particular, the 3D T1-WI used a spoiled-gradient recalled (SPGR) protocol with an echo time (TE), an inversion time (TI), and a repetition time (TR) of 2.8 ms, 900 ms, and 5.9 ms, respectively; isotropic voxel of 1 × 1 × 1 mm^3^; field of view (FOV) of 25.6 × 25.6 cm^2^; flip angle (FLIP) of 10°. A 2D T2-FLAIR sequence was constructed using a TE, a TI, and a TR of 120 ms, 2100 ms, and 8500 ms; the same isotropic voxel size of 1 × 1 × 3 mm^3^; an FOV of 25.6 × 25.6 cm^2^; an FLIP of 90°. A 2D T1-WI spin echo (SE) was used with a TE/TR of 16 ms/600 ms, an FLIP of 90 degrees, an FOV of 25.6 × 25.6 cm^2^, and a voxel size of 1 × 1 × 3 mm^3^. Lastly, the DSC–PWI sequence was acquired during and after 15 mL of 0.1 mM/kg gadolinium-diethylenetriamine penta-acetic acid was injected with a power injector at a speed of 5 mL/s. It utilized single-shot echo-planar imaging with a TR/TE of 2275 ms/45 ms, an FLIP of 90°, an FOV of 25.6 × 25.6 cm^2^, an echo train length of 1, a bandwidth of 250 kHz, and a voxel size of 2 × 2 × 4 mm^3^. A total of 40 volumes were acquired.

A semi-automated thresholding/contouring method was used to determine the T1 and T2 lesion volumes (LV) on the T1-WI and T2-FLAIR sequences, respectively. Brain volumes for the whole brain (WB), white matter (WM), gray matter (GM), deep GM (DGM), and thalamus were determined with 3D T1-WI using cross-sectional Structural Image Evaluation, using Normalisation of Atrophy (SIENAX) and FMRIB’s Integrated Registration and Segmentation Tool (FIRST) protocols [17]. To prevent tissue misclassification, T1 hypointensities were inpainted before segmentation [18]. By excluding the regions of interest derived from the T2 LV map, volumes of the normal-appearing (NA) WB and NAWM were also produced.

The Java Image Manipulation (JIM) Perfusion Toolkit (Xinapse Systems, version 6.0, Essex, U.K.) was utilized for calculating the PWI-derived cerebral blood volume (CBV), cerebral blood flow (CBF), and mean transit time (MTT). This procedure included a motion correction and automatic identification of the arterial input function (AIF) before deriving the CBV, CBF, and MTT maps. The selected AIF voxels were manually inspected to ensure accurate identification. The structural and perfusion-based segmentations were co-registered in the same MRI space. The mean values of the CBV, CBF, and MTT measures for the brain regions were calculated. The MTT was taken as an absolute measure in seconds (s), whereas the CBF and CBV were relative and unique for each patient as we did not have a method to quantify the T1 changes. Thus, the CBV and CBF measures were divided by the corresponding NAWM value to obtain the normalized CBV (nCBV) and CBF (nCBF) [19].

### 2.3. Serum Neurofilament Light Chain Analysis

Blood samples were collected and appropriately stored until the batch analysis. Anticoagulant ethylenediaminetetraacetic acid (EDTA) vials were used for the collection of blood samples (100 µL of volume) and stored at −80 °C without a prior thaw cycle. The sNfL levels were derived with a validated single-molecule array assay (Simoa, Quanterix Corporation, Lexington, MA, USA) and quantified in picograms per milliliter. The assay analytical sensitivity was at 0.32 pg/mL, with the coefficient of variation below 8%. All assay analyses were conducted at University Hospital, Basel, Switzerland as part of a larger collaborative project [20]. A description and the validation of the Simoa assay can be found elsewhere [21].

### 2.4. Statistical Analyses

Statistical analyses were performed using SPSS version 26.0 (IBM, Armonk, NY, USA). The data distributions of the demographic, clinical, MRI, and serum-based variables were assessed using the Kolmogorov–Smirnov test for normality. Data with normal and non-normal distributions were described as either mean (standard deviation (SD)) or median (interquartile range (IQR)), respectively. Age- and sex-adjusted multivariable linear regression models assessed the association between sNfL and PWI-based measures (MTT, nCBV, and nCBF). The models were built with two blocks; first, a enter-based step was used, which adjusted for the main effects of age and sex, and second, a stepwise-based step was used, which included perfusion measures if they significantly explained the greater sNfL level variance (significant *R*^2^ change with the stepping method criteria with an entry *F* probability of 0.05 and removal of 0.1). Regression model variables of *R*^2^, standard error of estimate, *t*-statistics, standardized *β*, and *p*-values are reported. This type of analysis allowed for the identification of PWI predictors that would independently explain the additional variance in sNfL levels while correcting for the known effect of age. A false discovery rate (FDR) correction for multiple comparisons was performed using a Benjamini–Hochberg (BH) procedure. Post-hoc analyses included measures of the whole brain volume (WBV) and thalamic volume as adjusting variables in either the stepwise block or as variables in the initial adjusting block. Models with additional corrections for BMI and DMT were performed. Corrected *p*-values lower than 0.05 were considered statistically significant.

## 3. Results

### 3.1. Patient Characteristics

The sample consisted of 86 MS patients (64 females, 74.4%), with an average age of 54.1 years, disease duration of 20.8 years, and median disability level of 2.5 on the Expanded Disability Status Scale (EDSS). In terms of subtypes, five patients were classified as having CIS, 51 as having RRMS, and 30 as having PMS (25 SPMS and 5 PPMS). Furthermore, the MS patients had an average BMI of 27.3 and a median sNfL level of 24.3 pg/mL. In terms of DMT, the MS patients were treated with interferon-β (*n* = 24, 27.9%), glatiramer acetate (*n* = 26, 30.2%), natalizumab (*n* = 3, 3.5%), oral medications (fingolimod *n* = 3, 3.5%; teriflunomide *n* = 3, 3.5%; dimethyl fumarate *n* = 1, 1.2%), off-label medications (rituximab, intravenous immunoglobulins, methotrexate, mitoxantrone, and azathioprine, each *n* = 1, 1.2%), and 21 were not on any medication (24.4%). Twenty-seven MS patients (31.4%) had at least one relapse in the past 5 years and the overall 5-year annualized relapse rate (ARR) was 0.166.

All DSC–PWI measures are shown in Table 1. The MTT ranged from the longest 3.52 s in the NAWM to the shortest with 3.18 s in the thalamus. Similarly, the nCBV and nCBF ratios ranged from 1.8 and 1.9 in the GM to 1.5 and 1.7 in the thalamus. The thalamic PWI measures were relatively lower in comparison to the total DGM region. Similar descriptive data regarding the CIS/RRMS and PMS subtypes are shown in Table 2. PMS patients had a significantly longer thalamic MTT when compared to the CIS/RRMS group (3.04 vs. 3.42 s, *p* = 0.049).

### 3.2. Associations between sNfL and DSC–PWI-Based Measures

The regression models showing the age- and sex-adjusted associations between sNfL and PWI-based measures are presented in Table 3. In addition to the sNfL variance explained by age and sex (*R*^2^ = 0.329), both the thalamic and NAWM MTT measures provided a significant increase in *R*^2^ to 0.369 (*p* = 0.005) and 0.402 (*p* = 0.037), respectively. In particular, a longer thalamic MTT time (indicative of lower perfusion) was associated with greater sNfL levels (standardized *β* = 0.648, *t*-statistics = 2.868, adjusted *p*-value = 0.011). (Figure 1) The effect of the NAWM MTT did not survive the multiple comparison correction (BH-adjusted *p*-value = 0.053). In the post-hoc analysis, the thalamic MTT remained a significant predictor of the sNfL level variance (standardized *β* = 0.679, *t*-statistics = 2.906, *p*-value = 0.005), whereas the WBV and thalamic volume were excluded. Moreover, when the thalamic volume was adjusted in the first block, the thalamic MTT still remained a significant predictor of the sNfL level variance (standardized *β* = 0.207, *t*-statistics = 2.202, adjusted *p*-value = 0.031).

Similar findings were seen in the regression models for nCBV and nCBF with increases in *R*^2^ to 0.378 and 0.447, respectively (Table 3). In addition to age and sex effects, a lower nCBV of the thalamus was associated with greater sNfL levels (standardized *β* = −0.221, *t*-statistics = −2.529, *p* = 0.013, adjusted *p*-value = 0.022). Correspondingly, a lower nCBF of the thalamus was also associated with greater sNfL levels (standardized *β* = −0.346, *t*-statistics = −4.188, *p* < 0.001, adjusted *p*-value = 0.001).

After further adjusting for the BMI, the DSC–PWI variables did not remain as significant factors in the analysis. The potential confounding effects of BMI on the PWI-based and sNfL measures are discussed further below. After adjusting only for the effects of age, sex, and DMT, the thalamic PWI-based measures remained statistically significant (*r* = 0.248, *p* = 0.025 for the MTT; *r* = −0.423, *p* < 0.001 for the nCBF; and *r* = −0.27, *p* = 0.014 for the nCBV).

Lastly, the sNfL levels were not associated with any PWI-based measures within the T1 and T2 lesions. In particular, the MTTs of the T1 and T2 LVs were not associated with the sNfL level (age- and sex-adjusted *r* = 0.236, *p* = 0.08 and *r* = 0.354, *p* = 0.104). Similarly, the sNfL levels were not associated with the T1 nCBV (*r* = 0.017, *p* = 0.903) and the T2 nCBV (*r* = −0.053, *p* = 0.638), or with the T1 nCBF (*r* = 0.075, *p* = 0.583) and the T2 nCBF (*r* = 0.021, *p* = 0.853).

## 4. Discussion

In addition to the age and sex effects, this cross-sectional biomarker study demonstrated that lower thalamic perfusion explained the significantly greater sNfL level variance. Although marginal, the PMS patients had significantly longer MTTs when compared to the CIS/RRMS counterparts.

Regardless of the PWI technique utilized, a recent systematic review has described a total of twelve different studies that investigate associations between clinical MS disability and perfusion metrics [22]. Seven out of the twelve studies describe positive findings, where three analyses demonstrated significant associations between a longer MTT and clinical disability scores (either the EDSS or the Multiple Sclerosis Severity Score) [23,24,25]. Corroborating our findings, these studies implicate pathology within the thalamus or the overall DGM region. These GM structures are considered to be major structural and functional connectivity hubs through which axons traverse and connect cortical regions [26]. Therefore, thalamic injury (through direct axonal transection, hypoperfusion, or neurodegeneration) would result in the release of free NfL into the CSF and serum. As an alternative explanation for the sNfL-perfusion association, we should also consider that the hypoperfusion may be a result of the lower energy/blood flow demand of previously damaged axons. However, sNfL levels are generally considered to be biomarkers with a relatively short temporal window in which changes relate to the acute state of axonal damage [5]. Conversely to our hypoperfusion association, these acute MS changes, commonly seen as contrast-enhancing MS lesions, are accompanied by inflammation-induced hyperperfusion [7]. Regardless of the proposed directionality in the casual relationship between hypoperfusion and sNfL levels, we demonstrated that both changes in perfusion and neurodegeneration may describe overlapping MS pathophysiological mechanisms. That said, without a longitudinal study, both explanations remain equally plausible.

After adjusting for the BMI and DMT, our models did not include any PWI measures as factors explaining the additional sNfL level variance. These findings can be explained by the fact that the BMI can significantly modulate both perfusion and sNfL measures and act as a mediator between them. Since sNfL levels are relatively proportional to the number of damaged axons, its serum concentration is directly dependent on the compartment denominator [27]. Therefore, sNfL levels are highly dependent on the total body blood volume, and thus, on the patient’s BMI (heavier patients with greater blood volume will have lower sNfL levels when compared to lighter patients with an equivalent pathology) [27]. On the other hand, the BMI significantly modulates the patient vessel size and the CBV/CBF [28]. A similar argument can be made for the model adjustment with DMTs. Traditionally, the type of medication is prescribed according to the disease activity and disability levels, which have been associated with both biomarkers utilized in this study. Therefore, adjusting for DMT would bridge and lower the significance of the correlation.

This study has several design limitations. First, our DSC–PWI acquisition did not allow for the calculation of absolute CBV and CBF measures. We attempted to mitigate this limitation by normalizing the cortical and thalamic perfusion data with the perfusion measure of the corresponding NAWM region. However, the degree of NAWM pathology can vary across individual MS patients and between the MS subtypes. Due to ethical considerations, the use of gadolinium-based contrast perfusion imaging additionally prevented the acquisition of such metrics in a healthy control population. Both limitations can be effectively bypassed via the future use of an ASL sequence. Lastly, the cross-sectional study design can only provide associations between the two identified disease biomarkers without any temporal connotation. Future longitudinal studies should determine whether the lower perfusion is due to already existing thalamic axonal pathology (lower need for perfusion) or whether the perfusion contributes to greater axonal damage (ischemia leading to neuronal loss).

In conclusion, decreased thalamic perfusion, measured as a longer MTT and a lower nCBV and nCBF, was associated with greater sNfL levels in a heterogeneous population of MS patients. In addition to age effects, perfusion measures explained the significantly greater sNfL level variance.

## Figures and Tables

**Figure 1 diagnostics-10-00685-f001:**
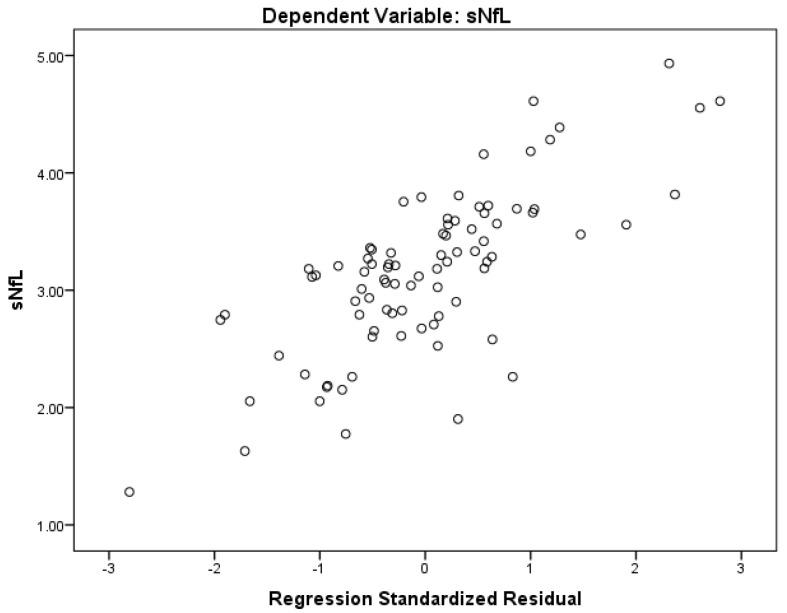
Example of regression-based associations between the sNfL level and the MTT-based model. sNfL: serum neurofilament light chain, MTT: mean transit time. The scatterplot demonstrates the association between the sNfL levels with the standardized residuals derived from the three regression predictors (age, sex, and thalamic mean transit time). This scatter plot is equivalent to an age- and sex-adjusted partial correlation between the sNfL level and thalamic MTT.

**Table 1 diagnostics-10-00685-t001:** PWI characteristics of the MS (*n* = 86) population.

ROI	MTT	nCBV	nCBF
NAWB	3.46 (0.74)	1.4 (0.08)	1.4 (0.13)
NAWM	3.52 (0.73)	-	-
GM	3.44 (0.75)	1.8 (0.18)	1.9 (0.32)
DGM	3.25 (0.78)	1.6 (0.14)	1.9 (0.29)
Thalamus	3.18 (0.85)	1.5 (0.17)	1.7 (0.29)

PWI: perfusion-weighted imaging, MS: multiple sclerosis, ROI: region of interest, NAWB: normal-appearing whole brain, NAWM: normal-appearing white matter, GM: gray matter, DGM: deep gray matter, MTT: mean transit time, nCBV: normalized cerebral blood volume, nCBF: normalized cerebral blood flow. SD: standard deviation. The values are given as mean (SD). The MTT is shown in seconds. nCBF and nCBV are unitless measures as they are the ratio relative to the corresponding region of the NAWM.

**Table 2 diagnostics-10-00685-t002:** Demographic, clinical, and PWI characteristics of the MS subgroups.

Demographic and Clinical Characteristics	CIS/RRMS (*n* = 56)	PMS (*n* = 30)	*p*-Value			
Female, *n* (%)	43 (76.8)	21 (70.0)	0.605			
Age, mean (SD)	49.6 (11.9)	62.5 (6.4)	**<0.001**			
BMI, mean (SD)	27.0 (4.9)	27.8 (5.6)	0.531			
Disease duration, mean (SD)	17.2 (9.9)	27.5 (10.3)	**<0.001**			
EDSS, median (IQR)	2.0 (1.0–4.0)	6.0 (3.5–6.0)	**<0.001**			
sNfL, median (IQR)	22.0 (13.3–27.9)	29.2 (21.8–44.6)	**<0.001**			
	**CIS/RRMS (*n* = 56)**			**PMS (*n* = 30)**		
**ROI**	**MTT**	**nCBV**	**nCBF**	**MTT**	**nCBV**	**nCBF**
NAWB	3.38 (0.69)	1.38 (0.07)	1.45 (0.15)	3.61 (0.81)	1.36 (0.09)	1.42 (0.87)
NAWM	3.47 (0.7)	-	-	3.62 (0.79)	-	-
GM	3.34 (0.69)	1.81 (0.16)	1.94 (0.34)	3.61 (0.84)	1.78 (0.22)	1.91 (0.2)
DGM	3.14 (0.7)	1.56 (0.17)	1.84 (0.29)	3.45 (0.89)	1.54 (0.15)	1.88 (0.38)
Thalamus	3.04 (0.73)	1.45 (0.15)	1.73 (0.31)	3.42 (1.0)	1.52 (0.17)	1.66 (0.25)

PWI: perfusion-weighted imaging, MS: multiple sclerosis, BMI: body mass index, CIS: clinically isolated syndrome, RRMS: relapsing-remitting MS, PMS: progressive MS, EDSS: Expanded Disability Status Scale, sNfL: serum neurofilament light chain, ROI: region of interest, NAWB: normal-appearing whole brain, NAWM: normal-appearing white matter, GM: gray matter, DGM: deep gray matter, MTT: mean transit time, nCBV: normalized cerebral blood volume, nCBF: normalized cerebral blood flow, SD: standard deviation, IQR: interquartile range. *χ*^2^, Student’s *t*-test, and the Mann–Whitney *U*-test were used for categorical, normally distributed numerical, and not-normally distributed numerical variables, respectively. *p*-values lower than 0.05 were considered statistically significant and shown as bold. The MTT is shown in seconds (s). nCBF and nCBV are shown as a ratio relative to the corresponding region of the NAWM.

**Table 3 diagnostics-10-00685-t003:** The linear regression models that were used to determine the associations between the sNfL levels using DSC–PWI-based measures.

**MTT**	***R*^2^**	**SE of the Estimate**	***t*-Statistic**	**Standardized *β***	***p*-Value**	**BH** ***p*-Value**
Block 1	0.329	0.568				
Sex			1.09	0.093	0.309	0.309
Age			5.338	0.479	<0.001	**<0.001**
Block 2						
Thalamus MTT	0.369	0.554	2.868	0.648	0.005	**0.011**
NAWM MTT	0.402	0.543	−2.119	−0.48	0.037	0.053
**nCBV**	***R^2^***	**SE of the Estimate**	***t*-Statistic**	**Standardized *β***	***p*-Value**	**BH** ***p*-Value**
Block 1	0.329	0.568				
Sex			1.118	0.098	0.267	0.297
Age			6.357	0.56	<0.001	**<0.001**
Block 2						
Thalamus nCBV	0.378	0.55	−2.529	−0.221	0.013	**0.022**
**nCBF**	***R^2^***	**SE of the Estimate**	***t*-Statistic**	**Standardized *β***	***p*-Value**	**BH** ***p*-Value**
Block 1	0.329	0.568				
Sex			1.389	0.115	0.168	0.211
Age			6.175	0.515	<0.001	**<0.001**
Block 2						
Thalamus nCBF	0.447	0.519	−4.188	−0.346	<0.001	**<0.001**

sNfL: serum neurofilament light chain, DSC–PWI: dynamic susceptibility contrast–perfusion-weighted imaging, MTT: mean transit time, NAWM: normal-appearing white matter, nCBV: normalized cerebral blood volume, nCBF: normalized cerebral blood flow, SE: standard error, BH: Benjamini–Hochberg. Each sNfL level linear regression model was built with a first block that force-entered and corrected for the effects of age and sex. The second stepwise block determined which PWI-based measure provided additional and significant explanatory power. The sNfL level was normalized using a natural logarithmic transformation. The false discovery rate (FDR) for multiple comparisons utilized the Benjamini–Hochberg procedure. Variables with a significant effect on the sNfL level are shown in bold.

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
