# Peer review of "Serum Neurofilament Light Chain Levels are Associated with Lower Thalamic Perfusion in Multiple Sclerosis"

_diagnostics, 2020, doi:10.3390/diagnostics10090685_

Round 1

Reviewer 1 Report

The manuscript entitled „Serum neurofilament light chain levels are associated with lower thalamic perfusion in multiple sclerosis” examines whether perfusion measures in MS are related or independent factors of disability in MS. The hypothesis was tested in a cross sectional study conducted on a heterogeneous MS population (n=86), which yielded two other publications in 2019. Greater age and thalamic MTT were independently associated with higher sNFL level.

Major points

  1. The MS population is heterogeneous. It is known from previous studies, that DMTs are influencing the actual sNFL level. Highly effective DMTs (such as natalizumab and fingolimod) are known to nearly normalize the NFL levels. The additional correction for DMTs were not shown in the manuscript. It is also stated in the paper, that after further adjusting for BMI and DMT effects, the DSC-PWI variables did not remain significant factors.

The patients number on highly effective IMTs were 7, the patients on moderately effective DMTs 58, until the number of patients untreated 21.

The course of the disease, whether active or inactive, remained unknown.

  1. The clinical relevance and pathogenesis of the brain perfusion MR imaging is still unclear (Lapointe et al. AJNR 2018), it is highly sensitive to inflammatory activity in the brain tissue. By requiring many technical steps, the technique is susceptible to measurement errors.
    What is the explanation to the statement for thalamic hypoperfusion and increased MS disabiltiy? Were other MR parameters than DSC-PWI related to sNFL (eg. T2 lesion load, etc.)? Were the lesions classified as active or chronic lesion?
  2. Correlation of age and higher sNFL is not a novelty. sNFL normalized for age can be used.

Minor points

  1. S1 R18 In the abstract “associated with MS” needs to be clarified
  2. S2 R50 The sentence starting with “Because of assay…” is hard to interpret.
  3. S2 R61 instead of “that only” was meant probably “that the only”
  4. Table 1: as the demographic data are described in the text, the content in the table just repeats it. I suggest that the table contains only the PWI data.
  5. Figure 1. needs interpretation, it is not self-explanatory
  6. S6 R232 The sentence starting with “Although we have mitigated…” is hard to understand
  7. spell check needed

Author Response

The manuscript entitled „Serum neurofilament light chain levels are associated with lower thalamic perfusion in multiple sclerosis” examines whether perfusion measures in MS are related or independent factors of disability in MS. The hypothesis was tested in a cross sectional study conducted on a heterogeneous MS population (n=86), which yielded two other publications in 2019. Greater age and thalamic MTT were independently associated with higher sNFL level.

Answer: We thank the Reviewer for the significant contribution and comments that improve the quality of our manuscript. Point-by-point answer to each comment are shown hereafter:

Major points                                                                                                

  1. The MS population is heterogeneous. It is known from previous studies, that DMTs are influencing the actual sNFL level. Highly effective DMTs (such as natalizumab and fingolimod) are known to nearly normalize the NFL levels. The additional correction for DMTs were not shown in the manuscript. It is also stated in the paper, that after further adjusting for BMI and DMT effects, the DSC-PWI variables did not remain significant factors.

The patients number on highly effective IMTs were 7, the patients on moderately effective DMTs 58, until the number of patients untreated 21.

Answer: The initial adjustment was for BMI and DMT combined. When there are no significant predictors included from the step-wise segment, a model is not created.

However, for the purpose of showing models only with DMT adjustment, we include the age, sex and DMT-adjusted partial correlations between sNfL and thalamic MTT, nCBV and nCBF (we report only the thalamic PWI-based measure from the last step of the step-wise model).

The course of the disease, whether active or inactive, remained unknown.

Answer: We included the 5-year annualized relapse rate and the 5-year relapse history of the MS participants. 27 participants had at least one relapse in the past 5 years, and the remaining 59 patients were clinically inactive.

 The clinical relevance and pathogenesis of the brain perfusion MR imaging is still unclear (Lapointe et al. AJNR 2018), it is highly sensitive to inflammatory activity in the brain tissue. By requiring many technical steps, the technique is susceptible to measurement errors. 
What is the explanation to the statement for thalamic hypoperfusion and increased MS disabiltiy? Were other MR parameters than DSC-PWI related to sNFL (eg. T2 lesion load, etc.)? Were the lesions classified as active or chronic lesion?

Answer: We thank the Reviewer for the suggestion to include lesion-based measures. The associations between PWI-based measured of the T2 lesion volume (LV) and T1-LV with sNfL levels are now in the Results section of the manuscript. Only 4 patients had presence of contrast-enhancing lesions (active) and this sample size was insufficient for statistical analysis.

Due to lack of longitudinal analysis based on Jacobian determinant, or susceptibility-weighted imaging (SWI) analysis for presence of iron-based ring, we were not able classify the lesions whether they were chronically active or not.

    2. Correlation of age and higher sNFL is not a novelty. sNFL normalized for age can be used.

Answer: We agree with the Reviewer. The association between sNfL and age is not novel finding and therefore was utilized as a mandatory correction step in all of our models. Whereas the first block of the regression models was used to correct for the significant age effect, the second block of the regression model only included variables that remain significant after that correction (therefore, all significant values are corrected for age and sex effects). We further clarified the statistical analysis and the age/sex correction in the manuscript.

 Minor points

  1. S1 R18 In the abstract “associated with MS” needs to be clarified

Answer: This was additionally corrected.

    2. S2 R50 The sentence starting with “Because of assay…” is hard to interpret.

Answer: This was additionally corrected.

    3. S2 R61 instead of “that only” was meant probably “that the only”

Answer: We thank the Review for the suggestion. We have updated the manuscript.

    4. Table 1: as the demographic data are described in the text, the content in the table just repeats it. I suggest that the table contains only the PWI data.

Answer: This was removed as the Reviewer has suggested.

    5. Figure 1. needs interpretation, it is not self-explanatory

Answer: We thank the Reviewer for the comment. This was further explained in the legend of the Figure 1.

    6. S6 R232 The sentence starting with “Although we have mitigated…” is hard to understand

Answer: We thank the Reviewer for the comments. We have separated the sentence in to shorted and clearer sentences.

    7. spell check needed

Answer: We carefully checked the entire manuscript for any spelling and grammatical errors.

Reviewer 2 Report

The manuscript submitted by Jakimovski and collaborators explore the associations between serum neurofilament light chain levels and lower thalamic perfusion in relapsing-remitting multiple sclerosis (RRMS), progressive (primary and secondary combined) MS, and clinically isolated syndrome (CIS) patients. Further, the analysis was performed taking into consideration the effects of age, sex, and body mass index. A total of 86 MS patients were included in the study. Studies of the association between PWI-based measures and serum neurofilament light chain is of relevance for the search for non-invasive biomarkers in the MS population. The manuscript is well written, easy to read, and informative, highlighting both strengths and limitations of the study, including the need for longer cross-sectional analyses and the comparison with measures obtained from healthy individuals under appropriate ethical standards. There are a few minor comments to consider:

- brain perfusion and its importance in MS pathology should be defined somewhere in the introduction.

- In the methods section, the preparation and the volume of serum used should be included.

- Also, the catalog number of the Simoa kit should be included.

-  If permitted, the supplemental figure 1 might be included as part of the manuscript

Author Response

The manuscript submitted by Jakimovski and collaborators explore the associations between serum neurofilament light chain levels and lower thalamic perfusion in relapsing-remitting multiple sclerosis (RRMS), progressive (primary and secondary combined) MS, and clinically isolated syndrome (CIS) patients. Further, the analysis was performed taking into consideration the effects of age, sex, and body mass index. A total of 86 MS patients were included in the study. Studies of the association between PWI-based measures and serum neurofilament light chain is of relevance for the search for non-invasive biomarkers in the MS population. The manuscript is well written, easy to read, and informative, highlighting both strengths and limitations of the study, including the need for longer cross-sectional analyses and the comparison with measures obtained from healthy individuals under appropriate ethical standards. There are a few minor comments to consider:

Answer: We thank the Reviewer for the suggestions that further improved the quality of our manuscript. Our point-by-point answers are provided hereafter:

- brain perfusion and its importance in MS pathology should be defined somewhere in the introduction.

Answer: This was further included in the Introduction.

- In the methods section, the preparation and the volume of serum used should be included.

- Also, the catalog number of the Simoa kit should be included.

Answer on both queries: We included additional information regarding the Simoa analysis (Quanterix Corporation, Lexington, MA).

-  If permitted, the supplemental figure 1 might be included as part of the manuscript

Answer: We have moved the supplementary table 1 into the manuscript.

Round 2

Reviewer 1 Report

Congratulations to the athors.

There is a typo in the legend of Table 2. figure legen: ratio instead of "ration"